# Biological Characterization of *Polystichum lonchitis* L. for Phytochemical and Pharmacological Activities in Swiss Albino Mice Model

**DOI:** 10.3390/plants12071455

**Published:** 2023-03-27

**Authors:** Jan Sher, Gul Jan, Muhammad Israr, Muhammad Irfan, Nighat Yousuf, Fazal Ullah, Abdur Rauf, Abdulrahman Alshammari, Metab Alharbi

**Affiliations:** 1Center for Integrative Conservation, Yunnan Key Laboratory for Conservation of Tropical Rainforest and Asian Elephant, Xishuangbanna Tropical Botanical Garden, Mengla 666303, China; 2University of Chinese Academy of Sciences, No. 19A Yuquan Road, Beijing 100049, China; 3Department of Botany, Garden Campus, Abdul Wali Khan University Mardan, Mardan 23200, Pakistan; 4Department of Botany, Government Post Graduate College Mardan, Mardan 23200, Pakistan; 5Department of Botany, University of Swabi, Swabi 23561, Pakistan; 6Missouri Botanical Garden, 4344 Shaw Blvd., St. Louis, MI 63110, USA; 7Key Laboratory of Engineering Plastic, Institute of Chemistry, Chinese Academy of Sciences, Beijing 100049, China; 8Department of Chemistry, University of Swabi, Swabi 23561, Pakistan; 9Department of Pharmacology and Toxicology, College of Pharmacy, King Saud University, P.O. Box 2455, Riyadh 11451, Saudi Arabia

**Keywords:** analgesic, anti-inflammatory, antipyretic, antispasmodic, antioxidant, phytochemistry, *Polystichum lonchitis*

## Abstract

*Polystichum lonchitis* L. is a fern belonging to the family Dryopteridaceae. The present study was conducted to evaluate its pharmacological, antioxidant, and phytochemical properties, and to conduct GC–MS screening of *P. lonchitis*. The acetic acid-induced writhing test, yeast-induced hyperpyrexia method, carrageenan-induced rat paw edema model, and charcoal meal test model were carried out to assess analgesic, antipyretic, anti-inflammatory, and antispasmodic activity, respectively. DPPH was used as an antioxidant, while the phytochemical screening was conducted using standard scientific methods. Among the pharmacological activities, the most significant effects were observed in the analgesic and anti-inflammatory activities, followed by the antipyretic and antispasmodic activities, at a dose of 450 mg/kg after the 4th hour, compared with 150 mg/kg and 300 mg/kg. For the evaluation of antioxidant activities, the most significant results were detected in the methanolic and aqueous extracts. The detection of flavonoids and phenol occurred most significantly in the methanolic extract, and then in the ethanolic and aqueous extracts. The main compounds detected using GC–MS analysis with a high metabolic rate was 𝛼-D-Galactopyranoside, which had a metabolic rate of 0.851, and methyl and n-hexadecanoic, which had a metabolic rate of 0.972. Overall, the results suggested that *P. lonchitis* had a strong potential for pharmacological activities. The suggested assessment provided a way to isolate the bioactive constituents and will help to provide new medicines with fewer side effects. Due to the fern’s effectiveness against various diseases, the results provide clear evidence that they also have the potential to cure various diseases.

## 1. Introduction

Medicinal plants have been widely used since ancient times due to their potential ability to alleviate or cure diseases [1,2]. Medicinal plants have been a vital source of both curative and preventive medical therapy preparations for human beings, and have been used for the extraction of important bioactive compounds [2,3,4]. The World Health Organization (WHO) has suggested that 80% of people prefer to use plant extracts to cure various diseases [3]. Medicinal plants have the potential to provide a way for pharmaceutical uses for human ailments owing to their active compounds and chemical constituents relevant to biological activities [4,5,6,7]. Pharmacology has grown over the years as a scientific discipline that describes the special effects of vigorous chemicals [6]. Pharmacological studies have an essential role in determining the effects of chemical agents upon physiological, universal, subcellular, or behavioral processes, focusing on the cure and prevention of different diseases [7,8]. Ferns are ancient flora in the plant kingdom, with current research suggesting that approximately 12,000 species exist globally [1,8,9].

Ferns have a large range of medicinal uses and have authentic potential to cure diseases; however, scientists have paid insufficient attention to their phytochemical analysis and biological activities, which remain comparatively unexplored in this flora [10]. Phytochemical investigation of five medicinal ferns showed significant effects that proved that ferns contain high levels of bioactive chemical compounds, which play an important role in the field of pharmacology [11]. Prior research has found that methanolic extract from the leaves of *Angiopteris evecta* (Giant fern) delivered significant pain reduction (analgesia) and reduced inflammation compared with controls [12]. Other studies on ferns have shown that they have pharmacological effects, many of which have been used for different treatments [13,14,15]. Johnson et al.’s [16] study of different extracts of *Sphaerostephanos unitus* (L.) Holttum showed that *S. unitus* had a high antioxidant and anti-inflammatory potential. Sher et al.’s [17] study on *Dryopteris blanfordii* suggests that ferns have a high potential for anti-inflammatory, analgesic, and antipyretic use. Over recent decades, rural societies, races, and traditions around the world have used plant components such as rhizomes, stems, leaves, pinnae, and spores in various ways to target different diseases, which has intermittently attracted the attention of researchers studying taxonomy, ecology, and fern distribution; however, their findings have not yet received sufficient medical approval [17,18]. New plant species with high medicinal importance have been added to the flora of Pakistan [19].

The aims of the current study are to provide scientific evidence for the analgesic, anti-inflammatory, antipyretic, and antispasmodic activities of methanolic extracts of *P. lonchitis* using an appropriate model on albino mice. Efforts have been made to research medicinally important pteridophytes and to properly recognize their useful features. An extensive review of the scientific literature on ferns reveals that ferns are used in the treatment of various diseases through their analgesic, antipyretic, and anti-inflammatory properties, but there is no study have been conducted on *P. lonchitis* in the context of its pharmacological and phytochemical aspects. Therefore, the current investigation explores the analgesic, antipyretic, anti-inflammatory, antispasmodic, and antioxidant activities of, and conducts phytochemical screening of, different extracts of *Polystichum lonchitis* L.

## 2. Results

### 2.1. Performance of Different Methanolic Extracts/Doses against Pharmacological Activities

#### 2.1.1. Methanolic Extract Performance against Analgesic Activity

The group 1 (control) was exposed to writhing, i.e., 21.4 ± 0.10 in 5 min. Group 2 with a significant ratio reduced the writhing to 4.4 ± 0.02 * in 5 min and showed a 25% inhibition using the standard drugs (Aspirin). In the remaining groups, the most significant reduction in writhing were 7 ± 0.24 * in 450 mg/kg dose and a 30% reduction, followed by group 3 and group 4 (Table 1).

#### 2.1.2. Methanolic Extract Performance against Anti-Inflammatory Action

The anti-inflammatory action in the methanolic fraction was examined under the varying doses, in which group 3 received 150 mg/kg, group 4 received 300 mg/kg, and group 5 received 450 mg/kg, but no significant results were obtained for up to 3 h of inhibition. However, after the 4 h, group 4 and group 5 showed significant inhibition, followed by group 3. The lowest inhibition was observed in group 4 after the 4 h (0.92 ± 0.03 *) and group 5 was (0.86 ± 0.02 *) (Table 2; Figure 1).

#### 2.1.3. Methanolic Extract Performance against Antipyretic Activity

To assess antipyretic activity, methanolic extract of *P. lonchitis* was applied at different doses, i.e., 150 mg/kg, 300 mg/kg, and 450 mg/kg. Of these three doses, the most significant reduction was observed under a dose of 300 mg/kg (group 4) and 450 mg/kg (group 5) after the 3 and 4 h (Table 3 & Figure 2), followed by a dose of 150 mg/kg (group 3). The standard drug paracetamol in group 2 showed a significant reduction after the 2, and 4 h relative to the methanolic dose.

#### 2.1.4. Methanolic Extract Performance against Antispasmodic Activity

In antispasmodic studies, the most significant performance was observed at a dose of 300 mg/kg (group 4) (33.1 ± 0.09 *), 59.56% charcoal meal transit and 450 mg/kg (group 5) showed (21.5 ± 0.05) and 49.76% charcoal meal transit, followed by a dose of 150 mg/kg (group 3) (28 ± 0.08 *) and 63.34% charcoal meal transit, compared with group 2 standard drug atropine (37.2 ± 0.03) and % inhibition 78.71% (Table 4).

### 2.2. Phytochemical Studies

Phytochemical (qualitative and quantitative) analysis of *P. lonchitis* was carried out using methanolic, ethanolic, and aqueous extracts.

#### 2.2.1. Qualitative Detection of Bioactive Compounds of *P. lonchitis* in Leaf and Rhizome

The phytochemical detection of *P. lonchitis* was performed using methanolic, ethanolic, and aqueous extracts of the leaves and rhizomes. The methanolic extract showed the presence of flavonoids, saponins, alkaloids, phenols, carbohydrates, and tannins, while quinine was absent in the leaf but present in the rhizome. Analysis of the ethanolic extract showed the presence of all the tested biochemicals in the leaves, with only quinine undetected in the rhizome. In the aqueous extract, only the saponins were not detected in either part (Table 5).

#### 2.2.2. Quantitative Assessment of Total Flavonoids and Phenolic Content

Screening was conducted for quantitative analysis of the content of flavonoids and phenols by studying the whole plant using methanolic, ethanolic, and aqueous extracts. The various extract fractions were compared with the standard equation of flavonoids in which the R^2^ = 0.9997 while the value of y = 0.0031x + 0.0159. The highest level of contents was found in the methanolic extract (5.26 ± 0.81 * µg/mL), followed by the ethanolic extract (4.61 ± 0.50 * µg/mL), and the aqueous extract, which had a low quantity of flavonoids (2.91 ± 0.65 µg/mL). Regarding the phenolic content, the methanolic extract showed a high quantity of contents (6.35 ± 0.581 * µg/mL), followed by the ethanolic extract (5.00 ± 0.988 * µg/mL), and the aqueous extract (3.676 ± 0.039 * µg/mL). The methanolic extract again proved its significant potential to provide a high quantity and quality of phenolic and flavonoid contents (Table 6).

### 2.3. Antioxidant Action

The antioxidant actions of *P. lonchitis* were tested using methanolic, ethanolic, and aqueous extracts.

#### Antioxidant Scavenging Action of the *P. lonchitis* at Different Concentrations

The antioxidant percentage potential in *P*. *lonchitis* was examined at different concentrations, i.e., 0.05 mg/mL, 1 mg/mL, and 1.5 mg/mL, and each concentration was examined in the methanolic, ethanolic, and aqueous extracts. The most significant potential was identified in the ethanolic extract, followed by the methanolic and aqueous extract (Table 7). At a concentration of 0.05 mg/mL, the highest potential was shown by the ethanolic extract with 59.7% (0.54 ± 0.010); the second most significant was the methanolic fraction with 52.9% (0.41 ± 0.04); and finally, the aqueous extract was 28.58% (0.84 ± 0.004). At a concentration of 1 mg/mL, the percentage potential was more significant in the ethanolic extract, with 70% potential and 0.42 ± 0.001 value, followed by the methanolic extract (51.08%, 0.48 ± 0.11), and the aqueous extract (39.7%, 0.74 ± 0.01). More significantly, in comparison with the two other concentrations, a 1.5 mg/mL concentration showed a significant result in the ethanolic extract with 79% potential and a 0.41 ± 0.07 value. However, in the methanolic extract, the potential was 59% with a 0.44 ± 0.159 value, and in the aqueous extract it was 43.9% (0.69 ± 0.001). The analysis thus showed that the ethanolic extract fraction had the potential to result in antioxidant activity (Table 7).

### 2.4. GC–MS Studies

Gas chromatography and mass spectrometry were examined in a methanolic fraction owing to the high potential for the extraction of active biochemicals. Distinctive chromatograms of the methanolic fractions resulted. The analysis identified 20 known compounds of various classes (Table 8, Figure 3).The main compounds detected in the GC–MS study included lupeol, L-proline, tetratriacontane, ethanolamine, 4-amino butyric acid, 3-hydroxybenzoate, glycerol, ribose, fructose, maleic acid, succinic acid, caryophyllene oxide, phytol, hexadecenoic acid, methyl ester, 𝛼-d-galactopyranoside, methyl, n-hexadecenoic acid, 9,12-octadecadienoic acid (Z,Z)-, methyl ester, 7-dehydrodiosgenin, and friedelan-3-one.

## 3. Discussion

The present study aimed to evaluate the analgesic, anti-inflammatory, antipyretic, and antispasmodic activities of methanolic extracts of *P. lonchitis* using mice models, while phytochemical and antioxidant activity were tested using methanolic, ethanolic, and aqueous extracts.

### 3.1. Performance of Methanolic Extract against Pharmacological Activities

We did not observe a significant effect of the *P. lonchitis* methanolic extract at a low dose from 1–2 h, but after the 3 and 4 h we observed a significant effect at high doses of 300 mg/kg and 450 mg/kg relative to 150 mg/kg. The results suggests that *P. lonchitis* has a potential against pharmacological activities, but due to its less significant effect it may be that *Polystichum* ferns might contain fewer secondary phytoconstituent substances than angiosperm plants. It is possible that the significant analgesic, antipyretic, and anti-inflammatory effects observed with this extract may be attributed to the flavonoid, alkaloid and tannin contents shown by [20,21]. Previous research by [22] suggested that in the absence of flavonoid content, there was no significant analgesic potential. Our study supports a large body of literature [23,24,25]. Many studies have shown that plants with a high flavonoid content have significant potential to cure specific diseases and provide analgesia; they can also reduce inflammation, have antibacterial activities, are anti-angiogenic, and can improve the treatment of cancer and allergic reactions [26,27,28].

Plant-based extraction plays a vital role in identifying new drugs and developing new analgesics, fever relief medicines, and anti-inflammatory drugs. Our findings reveal the positive aspects of secondary metabolites, especially quinine, tannins, phenols, alkaloids, and flavonoids. Examination using the protocol for our specimens suggests that the ferns have active ingredients of phenolic compounds with a high content of hydroxyl groups. The fractionation in the ethanol and methanol extracts suggested a high potential for antioxidant activities, which might be helpful to improve the treatment of inflammation caused by free radicals. In our findings, the analgesic potential was significant compared with the standard aspirin dose, so the results prove that the fraction of phyto-constituents have the potential to relive pain.

The results based on various doses indicate that the reduction of action compared with standard drugs was directly proportional to the increasing dose, i.e., it was proved to be precisely attributed to the active ingredient. According to the previous assessment of [29], analgesic efficiency is also attributed to reducing the prostate glands via inhibition of cyclooxygenase activities. As clearly evidenced, the analgesic action was mediated owing to the μ and κ receptors, although spinal pain was relieved by the δ κ μ opioid receptor [30,31]. Arachidonic acid is released from tissue through the cyclooxygenase pathway in response to inflammation, which provides feelings of peripheral pain [32].

Our results may be based on suppression of the opioid receptors in the methanolic fraction, or this may have been the result of effect on the central opioid receptors. The rectal temperature of mice was noted with the use of a digital thermometer before the provision of the methanolic extract, with the doses of 300 mg/kg and 450 mg/kg methanolic extract fractions significantly reducing body warming after a duration of two to three hours. The applied doses notably also reduced the body temperature more than paracetamol (Table 3 and Figure 2). Overall, after the 24th hour, rectal temperature was reduced and normalized compared with the standard drugs. The antipyretic action might be due to disruption of the prostaglandin synthesis; however, this pathway needs to be explored with an accurate mechanism [33]. This inhibition might be due to disruption of the peroxygenase stage or central inhibition which could result in the linkage of peripheral inflammation with the central inflammation of dinoprostone [34]. It may be predicted that the extracts’ antipyretic action prevents the inhibition of prostaglandin synthetase inside the hypothalamus, similar to the action of nonsteroidal anti-inflammatory drugs. However, there is no clear proof that *P. lonchitis* interferes with biosynthesis in the hypothalamus, although this possibility has been supported by some correlated studies in which *Dalbergia odorifera* and *Litsea glutinosa* extracts were shown to have the potential to inhibit prostaglandin production [35].

We observed that paw edema volume was reduced after the 3 and 4 h of methanolic extract injection, but these reductions were observed more in the high-concentration doses of methanolic extract fraction, such as 300 mg/kg and 450 mg/kg, than in the 150 mg/kg dose. The anti-inflammatory action of the most of the plants was also related to the existence of saponins [36], terpenoids [37], alkaloids [38], while glycosides were mentioned by [39], and tannins was indicated by [36]. However, the anti-inflammatory action of the selected plant in carrageenan-induced paw edema has potential owing to the presence of some active compounds, possibly including glycosides, flavonoids, alkaloids, saponins, and terpenoids. In antispasmodic studies, the most significant performance was observed at doses of 300 mg/kg and 450 mg/kg, followed by a dose of 150 mg/kg (compared with group 2, given the standard drug atropine). These findings confirm the local use of this plant as an antispasmodic and anti-diarrheic agent.

### 3.2. Antioxidant Performance, GC–MS Analysis and Phytochemical Screening of P. lonchitis

In terms of antioxidant activity, the most significant results were found in the ethanol extracts, with a concentration of 1 mg/mL, followed by the methanol extracts and the aqueous extracts. Our results support the findings of the study by [40], which used a methanol extract of *Polystichum aculeatum* found to have significant antioxidant properties with an IC50 value of 0.45 ± 0.02 µg/mL. *Athyrium filix-femina* (Lady fern) shows the strongest ABTS free radical scavenging activity (29.85 ± 1.39 µmol Trolox/g plant). Derivatives of tannins, such as phenolic compounds, were considered to have high potential as antioxidants and free radical scavengers [41,42]. Like phenolic compounds, flavonoids also have the potential for antioxidant activities, in addition to their potential use against allergies and their capability to reduce inflammation [1]. It is well known that phenolic compounds have biological activities used in areas such as antioxidants, antidiabetic drugs, hepatoprotective drugs, anti-inflammatory drugs, antimicrobial drugs, and anticancer drugs [43,44,45,46,47]. The existence of components as analyzed by GC–MS analysis also reinforced the plant’s anti-inflammatory, analgesic, and antipyretic potential.

The major compounds included lupeol which possess anti-inflammatory potential [14,15,28] and anticancer potential [48]. L-proline levels in plants are administered by biosynthesis, catabolism, and transport among cells and various compartments of the cells [49]. Tetratriacontane possesses antibacterial and antifungal activities [50], ethanolamine and the ethanolamine compound have inhibitory potential as anti-inflammatory agents [51], while 4-aminobutyric acid has potential against antioxidant and anti-inflammatory uses [52]. Regarding 3-hydroxybenzoate, several biological activities of this compound have been reported. For example, [53] reported that phenolic acid such as 3-hydroxybenzoic acid possessed good antioxidant activity. P-hydroxybenzoic acid has also been known for its anti-inflammatory activity as reported by [54]; this compound may have also contributed to the analgesic, anti-inflammatory, and antioxidant activities exhibited by *A. nobilis*. Work carried out by [55] on *Moringa oleifera* stems showed that glycerol had antioxidant and anti-inflammatory properties. Ribose has been suggested for use in the management of congestive heart failure as well as other forms of heart disease and for chronic fatigue syndrome [56], while fructose, maleic acid and succinic acid compounds have anti-inflammatory potential [57]. Caryophyllene oxide is known to possess anticarcinogenic, anti-inflammatory, and antibacterial properties [58]. Phytol is an important compound reported to have antioxidant, cytotoxic, and antimicrobial properties [59]; hexadecanoic acid, methyl ester, a major phytoconstituent, is known to possess strong antimicrobial activity [60]; 𝛼-d-galactopyranoside, methyl possesses anti-inflammatory, antioxidant, and antidiabetic properties [61]; n-hexadecanoic acid possesses anti-inflammatory and antioxidant properties [62]; 9,12-octadecadienoic acid (Z,Z)-, methyl ester possesses anti-cancer, anti-inflammatory, and antiandrogenic properties [63]; 7-dehydrodiosgenin has been reported to have antibacterial, antifungal, cytotoxic, and antioxidant properties [64]; and Friedelan-3-one has been reported as having antibacterial, antifungal, anti-inflammatory, analgesic, antipyretic, and antihypertensive properties [65]. All the phytochemicals were found to be present, with only quinine absent in the methanolic extract, and carbohydrates and saponins absent in the ethanolic and aqueous extracts.

Previous assessment was carried by [43] on five fern specimens of the family Pteridaceae, and all the assessed ferns were found to have the active ingredients. The high level of flavonoid and phenol content was most elevated in the methanolic extract, followed by the ethanolic extract and the aqueous extract. Moreover, [66] used ethanolic, hydro-alcoholic, and aqueous extracts of seeds of *Peganum harmala*. The maximum number of flavonoids was found to be in the ethanolic extract (7.25 ± 0.50), while the hydro-alcoholic extracts showed a high amount of phenol (4.20 ± 0.318).

## 4. Materials and Methods

### 4.1. Plant Collection and Identification

*Polystichum lonchitis* L. were collected from the Khyber Pakhtunkhwa, Pakistan. The plant species were identified with the help of the literature [67,68,69,70], and a taxonomist, and were then transported to the herbarium of the Abdul Wali Khan University Mardan, Khyber Pakhtunkhwa, Pakistan for use in the study.

### 4.2. Drying and Powdering

The collected plant specimens were cleaned of soil and any damaged parts were removed, and then shade dried by machine for 72 h. The dried specimens were powdered with the help of an electric grinder, and the powdered materials were stored in an airtight bottle for the experiments [17].

### 4.3. Preparation of Extracts

The ashes powder was used for obtaining the fractions layers, and the extraction was performed with methanol, ethanol, and aqua with a continuous method, respectively, for 48 h. Each extract was filtered, and further evaporation was conducted for the obtaining of dry extracts using previous protocols with little modification [14,15].

### 4.4. Animals Used

Swiss albino mice (20–30 g) of 40 days age and male were selected for the experiment. The mice were provided by the National Institute of Health, Islamabad, Pakistan (NIH) with ethical committee approval No-07(AWKUM 07-03-2017). The mice were kept at a temperature of 20 to 25 C for 5 days in polypropylene cages and given standard foods provided by NIH.

### 4.5. Analgesic Activity In Vivo

#### Acetic Acid-Induced Writhing Method

The tests were performed according to standard methods [71]. Writhing was counted after 20–25 min by injection of 1 mL acetic acid to all mice via the abdomen. The mice were divided into 5 groups. There was a total of 3 replicates for each treatment dose except for the control group, in which we used only one mouse, while for the remaining groups we used a total of 3 replicates: 3 mice were taken for the standard drug (group 2), 3 mice for the 150 mg/kg dose (group 3), 3 mice for the 300 mg/kg (group 4), and 3 mice for the 450 mg/kg (group 5). Group 1 served as a control and received only normal saline (1% *v*/*v*), group 2 received standard drugs (Aspirin) and for the remaining groups, group 3 received 150 mg/kg, group 4 received 150 mg/kg, and group 5 received 450 mg/kg doses, respectively. The percentage inhibition of analgesic activity was determined by the following formula: $$\%\;\mathrm{inhibition}=\frac{Number\;\;of\;writhings\;in\;tested\;drug}{Number\;of\;writhings\;in\;control}\times 100$$

### 4.6. Anti-Inflammatory Activity

The carrageenan-induced paw edema test model was used for anti-inflammatory activity. The mice were divided into 5 groups. There as a total of 3 replicates for each treatment dose, except for the control group, for which we used only 1 mouse and for the remaining group we used a total of 3 replicates: 3 mice were taken for the standard drug (group 2), 3 mice for the 150 mg/kg dose (group 3), 3 mice for the 300 mg/kg (group 4) and 3 mice for the 450 mg/kg (group 5). For the control group, the physiological saline was injected with a volume of 1% *v*/*v*, and the remaining 3 groups were injected with 1 mL carrageenan to cause paw edema. After the injection of the carrageenan, 20 min later the standard drug (Diclofenac sodium) was injected into group 2 and the methanolic extract at different doses of 150, 300, and 450 mg/kg were injected into group 3, group 4, and group 5, respectively. The paw edema reduction of mice was compared with the applied standard dose of group 2.The percentage inhibition of each applied dose was calculated with the help of the following formula [17,72].
$$\%\;inhibition=\frac{\left(Ct-Co\right)control-\left(Ct-Co\right)treated}{\left(Ct-Co\right)control}\times 100$$

*Ct* = Paw thickness at time *t*, *Co* = Paw thickness before carrageenan injection, *Ct* − *Co* = Paw edema, (*Ct − Co*) control = Paw edema after carrageenin injection to control group mice at time T, (*Ct* −*Co*) treated = edema after carrageenin injection to treated mice at time *t*.

### 4.7. Antipyretic Activity

The brewer’s yeast-induced pyrexia test model [73] was used for the antipyretic activity. Five groups of mice were taken with 3 replicates for each treatment except for the control group: 3 mice were taken for the standard drug (group 2), 3 mice for the 150 mg/kg dose (group 3), 3 mice for the 300 mg/kg (group 4) and 3 mice for the 450 mg/kg (group 5). Before the injection of the yeast, the initial rectal temperature of all the mice was measured. The group 1 mice (control group) was injected with 1 mL normal saline (NS) and the remaining groups, i.e., the 2nd, 3rd, 4th, and 5th groups, were injected with 1 mL brewer’s yeast. After the injection of the brewer’s yeast, the mice were left for 18 hours (h) and rectal temperature was then measured again of all the mice in all the groups. The group 2 mice received an injection of the standard drug (Paracetamol) and the remaining groups were injected with a fraction of the methanol extracts. Of these, group 3 received 150 mg/kg, group 4 received 300 mg/kg, and group 5 received 450 mg/kg doses, respectively. The rectal temperature of all the mice from group 1 to group 5 was checked after 1 h, 2 h, 3 h, and 4 h. The percentage reductions in antipyretic activity were determined by the following formula:% Reduction=B – Cn/B – A×100
where “*B*” represents temperature after pyrexia induction; *C_n_* represents the temperatures after 1, 2, 3, and 4 h; and “*A*” represents normal body temperature.

### 4.8. Antispasmodic Activity

Gastrointestinal motility using the charcoal meal test method was performed to test antispasmodic activity [74]. The mice were divided into 5 groups. There was a total of 3 replicates for each treatment dose except for the control group which we used only 1 mouse: for the remaining groups we used total 3 replicates as follows: 3 mice were taken for the standard drug (group 2), 3 mice for the 150 mg/kg dose (group 3), 3 mice for the 300 mg/kg (group 4) and 3 mice for the 450 mg/kg (group 5). Group 1 was used as a control and received a normal saline dose of 10 mL. The remaining groups of mice were given charcoal orally, and 30 min later the group 2 mice received the standard drug (Atropine sulfate), while the remaining groups of mice received methanolic extract orally at different doses as follows: group 3 received 150 mg/kg, group 4 received 300 mg/kg, and group 5 received 450 mg/kg doses. After the drug administration, the mice were sacrificed and the total charcoal meal length in the intestine and total intestine length were measured. The distances traveled by the charcoal meal in the intestine were estimated in terms of percentage of the applied doses. The percentage of intestinal transit was calculated using the formula:$$\left(\%\right)\;of\;Intestinal\;Transit=\frac{D}{L}\times 100$$
where “*D*” indicates the length in cm of (charcoal meal); and “*L*” indicates the total length in cm (intestine).

### 4.9. Antioxidant Activity

#### DPPH Radical Scavenging Method

The free radical scavenging activity was conducted according to the procedure of [75]. Different concentrations of the methanol, ethanol, and aqueous extracts of *P. lonchitis* were used for the antioxidant activity. During the experiment, DPPH (1,1-diphenyl-2-picrylhydrazyl) was used, which produced a violet and purple color using the methanol solution, and antioxidant activities were examined by the fade to shades of yellow. The absorbance of the mixtures was measured using a spectrophotometer at 517 nm with solvent and DPPH used as blanks.

### 4.10. Gas Chromatography–Mass Spectrometry Analysis

GC–MS analysis was conducted using a Clarus 600 gas chromatograph system equipped with a Clarus 600 C mass spectrometer (PerkinElmer Precisely, Waltham, MA, USA). An Elite-5MS fused silica capillary column coated with a 5% diphenyl/95% dimethylpolysiloxane stationary phase (60 m × 0.25 mm, film thickness 0.10 μm; Perkin Elmer Precisely, Cary, CA, USA) was used for gas chromatography. The injector temperature was maintained at 200 C whereas the oven temperature was increased from 80 C to 300 C for a total run time of 45 min. Inert gas helium was used as carrier gas at a flow rate of 1.0 mL/min. Initially, ethanol (blank) only was run from which the solvent delay was fixed to 4 min. The electron ionization mode with ionization energy of 70 eV; ion source temperature of 200 °C; GC interface temperature of 250 °C; scan interval of 0.2 s, and fragments range from 70 to 550 *m*/*z* were set for the mass spectrometry analysis. Manual injection of 1 μL of the ethanol extract of *P. lonchitis* in a spitless mode was followed [76].

### 4.11. Phytochemical Analysis

The required quantity of dry extracts was used for the qualitative phytochemicals analysis for compounds which included carbohydrates, alkaloids, flavonoids, phenols, saponins, quinine, and tannins in the methanolic, ethanolic and aqueous extracts of *P. lonchitis* using the methods of [11,12].

#### 4.11.1. Flavonoid Content Determination

Flavonoids are among the active constituents in plants, but the content extraction of flavonoids requires a proper protocol. The content was analyzed in various methanol, ethanol, and aqueous fractions. For each fraction, 4 mL was placed in the test tube, and 1 mL of potassium acetate and aluminum chloride was added. The volume thus reached 5 mL. It was then shaken for a period. The determination of absorbance was conducted by examining the 415 nm wavelength [77,78].

#### 4.11.2. Phenolic Content Determination

Phenols are also among the active ingredients of plants. In line with protocols, to the extracted content of plants, various fractions of methanol, ethanol, and aqueous solutions of 100 mL/L were added, and then from each stock solution 0.5 mL/L was transferred to the test tube, and 0.5 mL/L ethanol was added to each fraction solution. In addition, the solution was centrifuged at 10,000 rpm for 20 min. The supernatants were separated by centrifugation, and then all the tubes were equalized and centrifuged in line with previous protocols. Evaporation was then conducted with the help of a water bath. Finally, 3 mL/L distilled water and 0.5 mL/L follin-ciocalteu reagent was added to each test tube. The absorbance of content was checked with a 415 nm wavelength [79].

### 4.12. Statistical Analysis

For the statistical analyses, we used SPSS20 version IBM^®^, SPSS^®^ (New York, NY, USA). Each analysis was based on triplicates. The results used a mean ± standard error of mean (SEM) and were analyzed with statistical approaches by one-way analysis of variance (ANOVA). We also checked the data using the Tukey post-test. Regarding the value of significance, * *p* < 0.05 was considered significant [80].

## 5. Conclusions

The present study of pharmacological, antioxidant, and phytochemical properties of *P. lonchitis* extracts suggest they have a high potential against pharmacological activities, but the highest potential was observed in a high dose of methanolic extract, which could be attributed to the presence of bioactive agents including flavonoids, phenol, saponins, tannins, saponins, and alkaloids, suggesting that the chemical constitution of the plant could serve as a lead compound in the development of new analgesics and of anti-inflammatory, antipyretic, and antispasmodic agents.

These studies suggest future research directions to isolate the bioactive compounds of fern plants that may help to yield advanced medicines with fewer side effects and that are effective in their action against various diseases. Our results provide clear evidence that the fern has medicinal properties that could be used against different diseases.

## Figures and Tables

**Figure 1 plants-12-01455-f001:**
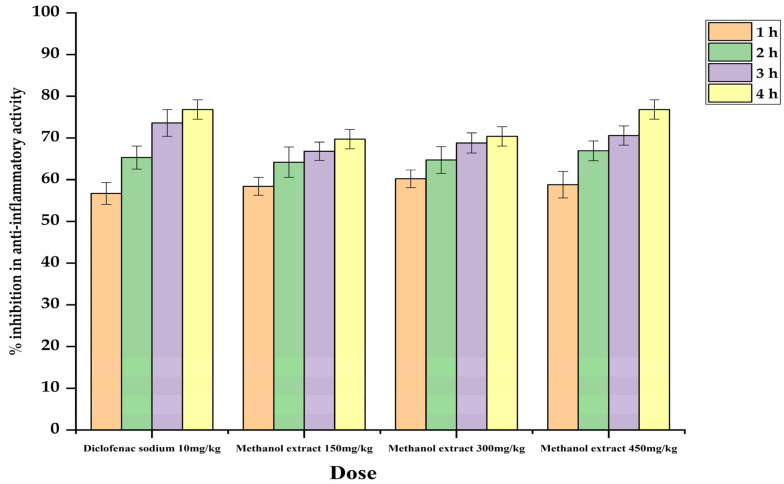
Percentage (%) inhibition in anti-inflammatory potential of different doses of methanolic extracts after 1 h, 2 h, 3 h and 4 h, respectively.

**Figure 2 plants-12-01455-f002:**
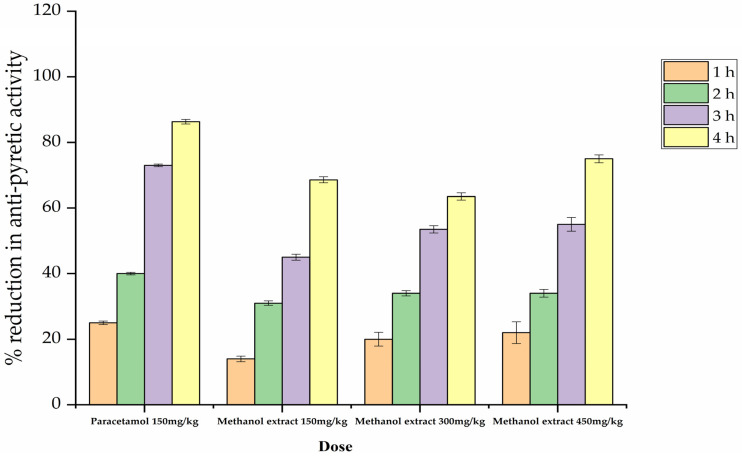
Percentage (%) reduction of antipyretic potential of different dose of methanolic extracts after 1 h, 2 h, 3 h and 4 h, respectively.

**Figure 3 plants-12-01455-f003:**
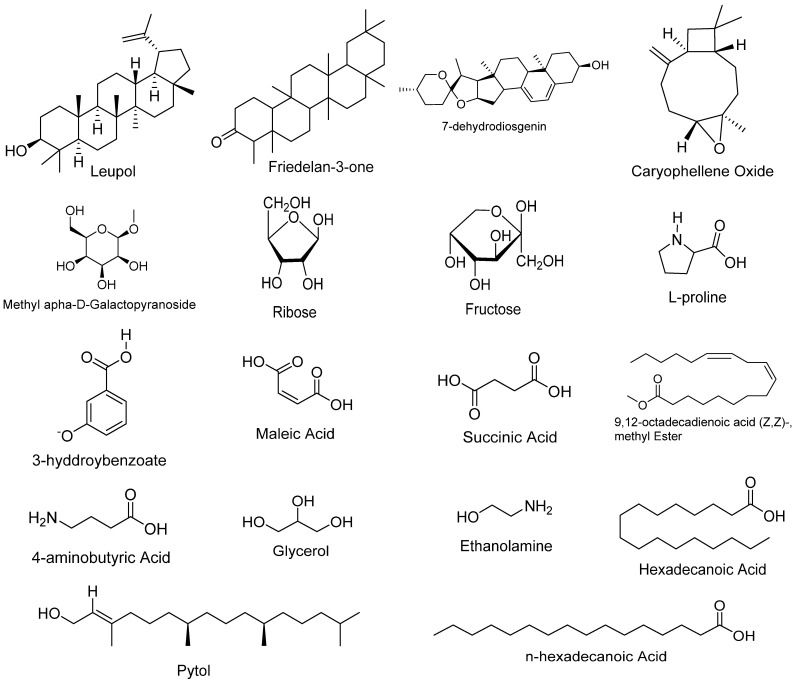
Molecular structures of identified compounds through GC–MS analysis.

**Table 1 plants-12-01455-t001:** Analgesic potential in methanolic extracts of different doses. All treatments consist of 5 groups. Each group has 3 replicates and was treated with different doses. The expression of value was mean ± SEM based. The mean difference was considered statistically significant at the 0.05 level. Significance indicated (* *p* < 0.05).

Groups	Dose	No of Writhing Rates in 5 min (Mean ± SEM)	% Inhibition of Mice
Group1 (control)	10 mL/kg (Normal Saline)	21.4 ± 0.10	----------
Group 2	150 mg/kg (Aspirin)	4.4 ± 0.02 *	25
Group 3	150 mg/kg	13 ± 0.07	60.74
Group 4	300 mg/kg	7.8 ± 0.05	36.44
Group 5	450 mg/kg	7 ± 0.24 *	30

**Table 2 plants-12-01455-t002:** Anti-inflammatory potential in methanolic extracts of different doses. All treatments consist of 5 groups. Each group has 3 replicates. Each group of mice was treated with a different methanolic extract/dose. The expression of value was mean ± SEM based. The mean difference was considered statistically significant at the 0.05 level. Significance indicated (* *p* < 0.05).

Treatment Groups	Dose	Paw Edema (mm^3^)		Paw Edema (mm^3^) after Drug Administration			
		Before Carrageenin Injection (Mean ± SEM)	1 h. after Carrageenin Injection (Mean ± SEM)	1 h (Mean ± SEM)	2 h (Mean ± SEM)	3 h (Mean ± SEM)	4 h (Mean ± SEM)
Group 1 (Control)		0.76 ± 0.05	1.72 ± 0.10	1.76 ± 0.08	1.93 ± 0.14	1.94 ± 0.0	1.98 ± 0.06
Group 2 (Diclofenac sodium)	10 mg/kg	0.83 ± 0.03 *	1.86 ± 0.12	1.48 ± 0.11	1.00 ± 0.11 *	0.25 ± 0.03 *	0.99 ± 0.02 *
Group 3	150 mg/kg	0.72 ± 0.05	1.14 ± 0.10	1.23 ± 0.12	1.33 ± 0.08	1.24 ± 0.06	1.53 ± 0.07
Group 4	300 mg/kg	0.78 ± 0.05	1.12 ± 0.06	1.10 ± 0.06	1.31 ± 0.06	0.98 ± 0.04 *	0.92 ± 0.03 *
Group 5	450 mg/kg	0.77 ± 0.05	1.86 ± 0.08	1.13 ± 0.06	1.11 ± 0.04 *	0.87 ± 0.03 *	0.86 ± 0.02 *

**Table 3 plants-12-01455-t003:** Antipyretic potential in methanolic extracts of different doses. All treatments consist of 5 groups. Each group has 3 replicates. Each group of mice was treated with a different methanolic extract/dose. The expression of value was mean ± SEM based. The mean difference was considered statistically significant at the 0.05 level. Significance indicated (* *p* < 0.05).

Treatment Groups	Dose	Rectal Temperature					
		Before Yeast Induction (Mean ± SEM)	After 18 h of Yeast Induction (Mean ± SEM)	1 h (Mean ± SEM)	2 h (Mean ± SEM)	3 h (Mean ± SEM)	4 h (Mean ± SEM)
Group 1 (control)	10 mL/kg	37.6 ± 0.14	39.1 ± 0.06	39.1 ± 0.07	39.1 ± 0.08	38.1 ± 0.06	38.0 ± 0.11
Group 2 Paracetamol	150 mg/kg	37.2 ± 0.14	39 ± 0.12	38.8 ±0.09	37.8 ± 0.04 *	37.1 ± 0.03 *	37.0 ± 0.03 *
Group 3	150 mg/kg	36.7 ± 0.02	38.7 ± 0.03	38.6 ± 0.08	38.4 ± 0.07	38.4 ± 0.09	38 ± 0.08
Group 4	300 mg/kg	37.2 ± 0.06	38.6 ± 0.06	38.7 ± 0.08	38.2 ± 0.06 *	37.9 ± 0.05 *	37.5 ± 0.05 *
Group 5	450 mg/kg	36.5 ± 0.02	38.9 ± 0.12	38.0 ± 0.07	38.2 ± 0.06	37.8 ± 0.04 *	37 ± 0.03 *

**Table 4 plants-12-01455-t004:** Antispasmodic potential in methanolic extracts of different doses. All treatments consist of 5 groups. Each group has 3 replicates. Each group of mice was treated with a different methanolic extract/dose. The expression of value was mean ± SEM based. The mean difference was considered statistically significant at the 0.05 level. Significance indicated (* *p* < 0.05).

Treatment Groups	Dose (mg/mL)	Total Intestine Length (cm) (Mean ± SEM)	Charcoal Meal Length (cm) (Mean ± SEM)	% Charcoal Meal Transit
Group 1 (control)		45.4 ± 0.1	_________	_______
Group 2 Atropine sulphate	10 mg/kg	45.4 ± 0.1	37.2 ± 0.03 *	78.71
Group 3	150 mg/kg	44.2 ± 0.25	28 ± 0.08 *	63.34
Group 4	300 mg/kg	55.6 ± 0.10	33.1 ± 0.09 *	59.56
Group 5	450 mg/kg	43.2 ± 0.33	21.5 ± 0.06	49.76

**Table 5 plants-12-01455-t005:** Qualitative detection of bioactive compounds of *P. lonchitis* leaves and rhizomes.

S.no	Phytochemical Test	Methanolic Extracts		Ethanolic Extracts		Aqueous Extracts	
		Leaf	Rhizome	Leaf	Rhizome	Leaf	Rhizome
1	Carbohydrates	+	+	+	+	+	+
2	Alkaloids	+	+	+	+	+	+
3	Flavonoids	+	+	+	+	+	+
4	Phenols	+	+	+	+	+	+
5	Saponins	+	+	+	+	−	−
6	Quinine	−	+	+	−	+	+
7	Tannins	+	+	+	+	+	+

(+) plus sign shows presence of and (−) negative sign indicates absence of phytochemical constituents.

**Table 6 plants-12-01455-t006:** Quantitative assessment of total flavonoids and phenolic contents. The mean difference was considered statistically significant at the 0.05 level. Significance indicated (* *p* < 0.05).

Plant Name	Part Used		Total Flavonoids Contents (µg/mL)	Total Phenolic Contents (µg/mL)
*P. lonchitis*		Extracts	Mean ± SEM	Mean ± SEM
	Whole	Methanol	5.26 ± 0.81 *	6.35 ± 0.581 *
	Whole	Ethanol	4.61 ± 0.50 *	5.00 ± 0.988 *
	whole	Aqueous	2.91 ± 0.65	3.676 ± 0.039 *

**Table 7 plants-12-01455-t007:** Antioxidant activity of *Polystichum lonchitis* L. The mean difference was considered statistically significant at the 0.05 level. Significance indicated (* *p* < 0.05).

Plant Part Used	Concentration	Extracts		Mean ± SD	% Potential
Whole	0.05 mg/ml	Methanol		0.41 ± 0.04 *	52.9
		Ethanol		0.54 ± 0.010 *	59.7
		Aqueous		0.84 ± 0.04 *	28.5
Whole	1 mg/ml	Methanol		0.48 ± 0.11 *	51.0
		Ethanol		0.42 ± 0.001 *	70
		Aqueous		0.74 ± 0.01 *	39.
Whole	1.5 mg/ml	Methanol		0.44 ± 0.01 *	65
		Ethanol		0.41 ± 0.07 *	79
		Aqueous		0.69 ± 0.08 *	43.9

**Table 8 plants-12-01455-t008:** Compounds identified through GC–MS of *P. lonchitis* whole plant and their metabolic rate.

Identified Compounds	Formula	Metabolic Rate
Lupeol	C_30_H_50_O	0.603
L-proline	C_5_H_9_NO_2_	0.587
Tetratriacontane	C_34_H_70_	0.472
Ethanolamine	C_2_H_7_NO	0.351
4-aminobutyric acid	C_4_H_9_NO_2_	0.582
3-hydroxybenzoate	C7H5O3-	0.721
Glycerol	C_3_H_8_O_3_	0.722
Ribose	C_5_H_10_O_5_	0.631
Fructose	C_6_H_12_O_6_	0.532
Maleic acid	C_4_H_4_O_4_	0.575
Succinic acid	C_4_H_6_O_4_	0.621
Caryophyllene oxide	C_15_H_24_O	0.721
Phytol	C_20_H_40_O	0.621
Hexadecanoic acid, methyl ester	C_17_H_34_O_2_	0.743
𝛼-D-Galactopyranoside, methyl	C_7_H_14_O_6_	0.851
n-hexadecanoic acid	C_16_H_32_O_2_	0.972
9,12-octadecadienoic acid (Z, Z)-, methyl ester	C_19_H_34_O_2_	0.534
7-dehydrodiosgenin	C_27_H_40_O_3_	0.432
friedelan-3-one	C_30_H_50_O	0.632

## Data Availability

Not applicable.

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
