# Peer review of "Biological Characterization of Polystichum lonchitis L. for Phytochemical and Pharmacological Activities in Swiss Albino Mice Model"

_plants, 2023, doi:10.3390/plants12071455_

Round 1
Reviewer 1 Report
Ref: Biological Characterization of Polystichum lonchitis L. for the Phytochemical and Pharmacological activities in Swiss albino mice model
The objectives of this study were to evaluate some biological properties of extracts from P. lonchitis. Some bioactive constituents were identified by GC-MS.
General aspects: The article is generally well written, containing the necessary information to understand the methodological procedures. Results are well discussed.
My considerations:
1. remove figures 1 and 4 as they are not needed. The data are already in table 1 and table 4, respectively.
2. discuss why lower inhibition values ​​were obtained for higher concentrations of the extracts in table 1
3. standardize the number of decimals in all tables
4. Discussion must be improved. It is long but confuse.
5. Add some of the key compounds identified in the Abstract
6. Add the units in values presented in lines 327 and 328
7. From my point of view, the main flaw of this work is that there was no attempt to correlate the biological activities with the chemical components. Discuss this point.
Author Response
RESPONSE:
We have attempted to address all reviewer points below. Here we provide a summary of the major changes we made: We would like to express our thanks to the reviewers for their thoughtful evaluations, which we believe have much improved the article. We uploaded the track version manuscript and pdf where we made changes, and we also uploaded clean version manuscript.
Reviewer 1:
Comments and Suggestions for Authors
Ref: Biological Characterization of Polystichum lonchitis L. for the Phytochemical and Pharmacological activities in Swiss albino mice model
The objectives of this study were to evaluate some biological properties of extracts from P. lonchitis. Some bioactive constituents were identified by GC-MS.
General aspects: The article is generally well written, containing the necessary information to understand the methodological procedures. Results are well discussed.
RESPONSE: thanks for your positive assessment.
My considerations:
- remove figures 1 and 4 as they are not needed. The data are already in table 1 and table 4, respectively.
RESPONSE: changed as suggested. Figure 1 and Figure 2 has been removed.
- discuss why lower inhibition values ​​were obtained for higher concentrations of the extracts in table 1
RESPONSE: This is according to the formula we mentioned in M$M part. The %inhibition value due to the methanolic extract. Under low concentrations of methanolic extract ie.150mg/kg, the number of writhing increased, if the number of writhing increased then the %inhibition increased. When the concentration increased then the % inhibition writhing decreased.
standardize the number of decimals in all tables
RESPONSE: changed as suggested.
- Discussion must be improved. It is long but confuse.
RESPONSE: we removed the repeating lines, I hope it will improved now
- Add some of the key compounds identified in the Abstract
RESPONSE: added as suggested.
- Add the units in values presented in lines 327 and 328
RESPONSE: We removed all values from the discussion parts as suggested by other reviewer. Now the value only in the tables.
- From my point of view, the main flaw of this work is that there was no attempt to correlate the biological activities with chemical components. Discuss this point.
RESPONSE: We attempted the biological activities only in crude extracts (methanolic extract). After the biological activities we found significant results such as after 3hrs and 4hrs in high dose 300 and 450mg/kg. Then we decided to do GC-MS to make sure that P. lonchitis plant contain compounds that have high metabolic rate and that can be use against inflammation, antipyretic, and analgesia, because this was the first study on P. lonchitis in the context of biological activities and phytochemicals screening. Through GC-MS analysis, we found some chemical compounds that can be use against inflammation, anti-pyretic and analgesic, for further detail please visit to discussion section. In discussion part, we further explained the functions of every compound.

Reviewer 2 Report
Introduction.
Row 46. The first sentence seems to be incomplete. Please revise.
Row 67-68. The authors mentioned “other studies” but they give only one example of studies, which shows the lack of background data.
Results
Row 96-97. Figure 1. The results presented here are redundant. The same data is presented in table 1. Moreover, the authors show some SEM bars in the figures, which looks like significant differences, however, in the figure this is not shown. Also, under Table 1 the P<0.05 is presented but there is no P value in the Table, only the SEM value is presented.
The same observation is for Figure 2 and Figure 3.
General observation. Under each table, the authors mention ‘G’ representing groups respectively’. It will be better to explain exactly what G1, G2, G3, G4 and G5 mean. Please revise.
Table 4 and Figure 4 also present repetitive information. It is not acceptable to present the results both in the table and in the figure as a chart. Chose one or the other way of presenting the results.
Moreover, the figure is not correctly presented. Needs revision.
As a general observation, my suggestion is to use a specific program to make all the graphs, not Microsoft excel.
Row 182-195. Table 8. The authors presented here the results of the Compounds identified by GC-MS, in the table, with a P value. This is wrong, the values should be given as average and SEM. It is impossible to obtain the P value from 3 averaged values.
General comment. In the entire results chapter, too much repetitive information. For example, the authors repeated the values from the tables, instead, they should indicate only the main findings, without making all these repetitions since the values are presented in the tables below.
Discussion section. Although the discussion part is well written, again the authors repeated the information presented in the table. Too much repetition of the values.
Row 345 – references not properly cited.
The entire material and Methods chapter is not cited in the order. Although the M&M was placed last in the paper, the order of the cited references should be continuous.
General comment, due to the small no of samples, describing a statistical model with P value is not quite accurate.
Author Response
RESPONSE:
We have attempted to address all reviewer points below. Here we provide a summary of the major changes we made: We would like to express our thanks to the reviewers for their thoughtful evaluations, which we believe have much improved the article. We uploaded the track version manuscript and pdf where we made changes, and we also uploaded clean version manuscript.
Reviewer 2:
Comments and Suggestions for Authors
Introduction.
Row 46. The first sentence seems to be incomplete. Please revise.
RESPONSE: We revised it, thank you for suggestions
Row 67-68. The authors mentioned “other studies” but they give only one example of studies, which shows the lack of background data.
RESPONSE: Changed as suggested. We have added 2 more examples, I hope it will be fine now.
Results
Row 96-97. Figure 1. The results presented here are redundant. The same data is presented in table 1. Moreover, the authors show some SEM bars in the figures, which looks like significant differences, however, in the figure this is not shown. Also, under Table 1 the P<0.05 is presented but there is no P value in the Table, only the SEM value is presented.
RESPONSE: We removed the Fig 1 and 4 because it was already presented in tables. We re-analyzed the data again due to missing of the (*P value). We have added the P value in all tables.
The same observation is for Figure 2 and Figure 3.
RESPONSE: We removed the Fig 1 and 4 because the %inhibition and reduction already explained in the tables. We improved Fig 2 and 3. For further please visit to clean version manuscript or track change, both files have been uploaded.
General observation. Under each table, the authors mention ‘G’ representing groups respectively’. It will be better to explain exactly what G1, G2, G3, G4 and G5 mean. Please revise.
RESPONSE: “G” was representing “group” in each table, we replaced “G” on group. So there is no need to explain it more.
Table 4 and Figure 4 also present repetitive information. It is not acceptable to present the results both in the table and in the figure as a chart. Chose one or the other way of presenting the results.
RESPONSE: We removed Fig1 and 4, so no need to discuss further
Moreover, the figure is not correctly presented. Needs revision.
RESPONSE: changes has been made as suggested by reviewer
As a general observation, my suggestion is to use a specific program to make all the graphs, not Microsoft excel.
RESPONSE: Yes, we used origin for all graphs
Row 182-195. Table 8. The authors presented here the results of the Compounds identified by GC-MS, in the table, with a P value. This is wrong, the values should be given as average and SEM. It is impossible to obtain the P value from 3 averaged values.
RESPONSE: Thank you for remind us. We removed P value because there is no need of P value in the table. We just shown the detective compounds through GS/MS and their metabolic rate. Now we make it simple and clear.
General comment. In the entire results chapter, too much repetitive information. For example, the authors repeated the values from the tables, instead, they should indicate only the main findings, without making all these repetitions since the values are presented in the tables below.
RESPONSE: We removed all the repetitive results, we explained only the main findings. I hope results will be Ok now.
Discussion section. Although the discussion part is well written, again the authors repeated the information presented in the table. Too much repetition of the values.
RESPONSE: We removed the repetition of results from discussion part. I hope discussion part is less repetitive and more accurate now.
Row 345 – references not properly cited.
RESPONSE: The references has been changed and arranged according to the Journal format.
The entire material and Methods chapter is not cited in the order. Although the M&M was placed last in the paper, the order of the cited references should be continuous.
RESPONSE: Changed as suggested
General comment, due to the small no of samples, describing a statistical model with P value is not quite accurate.
RESPONSE: I think 3 replicates are sufficient, the lowest/minimum sample can be three size/replicates but not less than three. With three biological replicates being the minimum for any inferential analysis. For the experiment, we tried to used 10-15 mice replicates for each group but we did not get sufficient mice for experiment, so we used only 3 replicates for each group.

Round 2
Reviewer 1 Report
After revision, the manuscript can be accept for publication
Reviewer 2 Report
No other observations.